# Multi-Omics Analysis of the Mechanism of Mentha Haplocalyx Briq on the Growth and Metabolic Regulation of Fattening Sheep

**DOI:** 10.3390/ani13223461

**Published:** 2023-11-09

**Authors:** Mingliang Yi, Zhikun Cao, Jialu Zhou, Yinghui Ling, Zijun Zhang, Hongguo Cao

**Affiliations:** 1College of Animal Science and Technology, Anhui Agricultural University, Hefei 230036, China; yimingliang@stu.ahau.edu.cn (M.Y.); 1471849743@stu.ahau.edu.cn (Z.C.); zhouhh@stu.ahau.edu.cn (J.Z.); lingyinghui@ahau.edu.cn (Y.L.); zhangzijun@ahau.edu.cn (Z.Z.); 2Anhui Province Key Laboratory of Local Livestock and Poultry Genetic Resource Conservation and Bio-Breeding, Anhui Agricultural University, Hefei 230036, China

**Keywords:** MHB, growth performance, microorganisms, metabolites, transcriptome

## Abstract

**Simple Summary:**

Mentha haplocalyx Briq (MHB) is a medicinal and edible herbal plant that has the effects of clearing heat, detoxifying, and aiding digestion as a traditional Chinese medicine. However, there is limited research on MHB as a feed ingredient for meat sheep. This study shows that adding MHB to feed can improve the growth performance of fattening sheep. We have revealed the mechanism by which MHB affects the growth and metabolism of meat sheep through aspects such as rumen and fecal microbial sequencing, rumen metabolomics, serum metabolomics, urine metabolomics, and rumen epithelial cell transcriptome sequencing. Our research results will provide a theoretical basis for the application of MHB in the production of meat sheep.

**Abstract:**

Mentha haplocalyx Briq (MHB) and its components have been proven to improve the growth performance of livestock and poultry. The aim of this experiment was to investigate the effects of MHB addition on growth performance, rumen and fecal microbiota, rumen fluid, serum and urine metabolism, and transcriptomics of rumen epithelial cells in meat sheep. Twelve Hu sheep were selected for the experiment and fed with basic diet (CON) and a basal diet supplemented with 80 g/kg DM of Mentha haplocalyx Briq (MHB). The experimental period was 10 weeks with the first 2 weeks as the pre-trial period. The results showed that compared with the CON group, the average daily weight gain of meat sheep in the MHB group increased by 20.1%; the total volatile fatty acid (VFA) concentration significantly increased (*p* < 0.05); The thickness of the cecal mucosal layer was significantly reduced (*p* < 0.01), while the thickness of the colonic mucosal layer was significantly increased (*p* < 0.05), the length of ileal villi significantly increased (*p* < 0.01), the thickness of colonic mucosal layer and rectal mucosal muscle layer significantly increased (*p* < 0.05), and the thickness of cecal mucosal layer significantly decreased (*p* < 0.05); The serum antioxidant capacity has increased. At the genus level, the addition of MHB changed the composition of rumen and fecal microbiota, increased the relative abundance of Paraprevotella, Alloprevotella, Marinilabilia, Saccharibacteria_genera_incertae_sedis, Subdivision5_genera_incertae_sedis and Ornatilinea in rumen microbiota, and decreased the relative abundance of Blautia (*p* < 0.05). The relative abundance of Prevotella, Clostridium XlVb and Parasutterella increased in fecal microbiota, while the relative abundance of Blautia and Coprococcus decreased (*p* < 0.05). There were significant differences in the concentrations of 105, 163, and 54 metabolites in the rumen, serum, and urine between the MHB group and the CON group (*p* < 0.05). The main metabolic pathways of the differences were pyrimidine metabolism, taurine and taurine metabolism, glyceride metabolism, and pentose phosphate pathway (*p* < 0.05), which had a significant impact on protein synthesis and energy metabolism. The transcriptome sequencing results showed that differentially expressed genes were mainly enriched in immune regulation, energy metabolism, and protein modification. Therefore, adding MHB improved the growth performance of lambs by altering rumen and intestinal microbiota, rumen, serum and urine metabolomics, and transcriptome.

## 1. Introduction

Healthy farming is the direction of current animal husbandry development, providing good growth and breeding conditions for animals through healthy farming, achieving healthy animal growth, and producing safe and high-quality animal products. Antibiotics or related drugs have been applied in animal husbandry production to reduce the incidence of animal diseases and improve the growth rate of livestock and poultry [1,2]. However, the abuse of antibiotics during the breeding process can lead to issues, such as dysbiosis of the microbial community, increased bacterial resistance, and decreased immunity in livestock and poultry [3]. At the same time, such livestock products can affect human health and public health safety through the food chain [4].

Mentha haplocalyx Briq (MHB), as an herbaceous plant, has functions such as clearing heat and detoxifying, being antibacterial and anti-inflammatory, enhancing appetite, and aiding digestion. It is a common ingredient in Chinese herbal medicine. MHB is native to China and widely distributed in the temperate regions of the Northern Hemisphere. China has the world’s largest cultivation area and yield of MHB [5,6]. In addition, MHB also has the characteristics of a fast growth rate and a high yield. Mint as a feed additive has the potential to promote poultry growth and improve feed conversion rates [7,8]. In the healthy breeding of quails, MHB and its derivatives have the ability to inhibit the toxic side effects of aflatoxin on the liver, bones, and meat, thereby improving the product quality of quails [9]. In addition, components such as phenols and flavonoids in MHB [10] have effects on reducing methane emissions from ruminants and regulating rumen fermentation [11]. MHB, as a forage resource, can be used for healthy breeding of meat sheep, achieving the goal of rational utilization of MHB forage resources while achieving healthy breeding of meat sheep.

However, there is still little research on the application of peppermint in healthy breeding of meat sheep. The use of antibiotics increases the economic cost of sheep farming and poses potential risks to humans. Therefore, MHB as a substitute for antibiotics is of great significance in achieving healthy sheep farming. In this study, MHB was added to the basic diet of fattening meat sheep. Through methods such as microbiology, metabolomics, and transcriptomics, the functions and related regulatory mechanisms of MHB in promoting growth, resisting oxidation, and regulating body metabolism during the fattening process of meat sheep were analyzed. The research results provide a scientific basis for the widespread application of MHB in healthy breeding of meat sheep and the use of antibiotics as a substitute.

## 2. Materials and Methods

### 2.1. Experimental Animals and Design

In this experiment, 12 Hu sheep (native Chinese sheep) were selected (body weights; BW, 23.0 ± 2.3 kg and ages 120 ± 3.5 d) with good physical condition. They were sourced from Zhoushi Sheep Industry Co., Ltd. in Chuzhou, China. The experimental plan has been approved by the Animal Ethics and Protection Committee of Anhui Agricultural University (NO: SYDW-P20190600601). Fattening sheep were randomly divided into 2 groups, with 6 replicates in each group. The control group (CON) was fed a basal diet, while the mint group (MHB) was fed a basal diet supplemented with 80 g/kg mint DM, with free intake and drinking water. The prefeeding period was 2 weeks, and the formal period was 8 weeks. The sheep shed maintained good ventilation, with appropriate temperature and humidity. The main nutritional indicators of the fattening sheep diet during the experiment are shown (Appendix A), and the experimental design and workflow are shown in the figure (Figure 1).

### 2.2. Collection and Determination of Samples

At the beginning and end of the experiment, the body weight (kg) of the fasting fattening sheep was measured, the average daily gain of the fattening sheep was calculated, and the feed intake of the fattening sheep was recorded during the experiment. The feed conversion rate is the average daily gain/average daily feed intake. One day before the end of the feeding experiment, fecal samples of 12 fattening sheep rectum were collected, with 30 g of each sample. In order to avoid environmental microbial contamination, the samples were quickly stored in liquid nitrogen for fecal microbial testing. Blood from 12 fattening sheep was collected from the jugular vein and allowed to stand at room temperature for 2 h. After the blood was completely agglutinated, it was centrifuged at 3000 r/min for 15 min. A light yellow liquid serum was taken and stored at −80 °C. The total antioxidant capacity (T-AOC) (A015-1-2, Nanjing Jiancheng Biotechnology Research Institute, Nanjing, China), superoxide dismutase (SOD) (A001-3-2, Nanjing Jiancheng Biotechnology Research Institute), glutathione peroxidase (GSH-Px) (A005-1-2, Nanjing Jiancheng Biotechnology Research Institute), catalase (CAT) activity (A007-1-1, Nanjing Jiancheng Biotechnology Research Institute), and malondialdehyde (MDA) content in serum (A003-1-2, Nanjing Jiancheng Biotechnology Research Institute) were measured using reagent kits. The remaining amount was used for metabolomics analysis. Fasting morning urine from 12 fattening sheep was taken and centrifuged in a 50 mL centrifuge tube at 4 °C at 3,000 rpm/min for 10 min to obtain the supernatant. The supernatant was then packaged in a 1.5 mL centrifuge tube, quenched in liquid nitrogen for 15 min and stored at 80 °C for metabolomics analysis. Once the feeding experiment was completed, the fasting fattening sheep were slaughtered and the rumen fluid samples collected. The rumen fluid samples were then filtered with 4 layers of gauze, and quickly stored in liquid nitrogen. The ammonia nitrogen concentration was measured by UV visible spectrophotometer through colorimetry, the volatile fatty acid concentration was measured by gas chromatography, and the remaining amount was used for microbiological and metabolomic analysis. Once the rumen epithelial tissue was collected, it was rinsed thoroughly with physiological saline and stored in an EP tube in liquid nitrogen for transcriptomic analysis. Various intestinal segments of fattened sheep were taken and fixed with 4% paraform-aldehyde. Histological sections were prepared using conventional HE staining, followed by fixation, dehydration, transparency, waxing, embedding, sectioning, staining, and sealing. The sections were then sealed with neutral gum and observed under a microscope. Throughout the entire sample collection process, disposable sterile gloves and masks were worn to avoid cross contamination. 

### 2.3. 16S rRNA Sequencing of Rumen Fluid and Fecal Microorganisms

The rumen fluid and feces were thawed at 4 °C, and the total bacterial DNA was extracted by E.Z.N.A. fecal DNA kit (Omega Bio tek, Norcross, GA, USA). The purity and concentration of the total DNA (OD260/280 and OD260/230) were detected by micro spectrophotometer, and the DNA quality was detected by 1% agarose gel electrophoresis. Primers 515F (5′-CCTACACGCTCTTCCGATCTN-3′) and 806R (5′-GACTGGATCCTGGCACCCGAGAATTCCCA-3′) were used to amplify the V3-V4 variable region of bacterial 16S rRNA. The PCR amplification reaction system was 30 µL: 15 µL 2× Phanta Master Mix; 1 µL Bar PCR prime F (10 µM), 1 µL prime R (10 µM), 10 ng Genomic DNA, supplemented with ddH_2_O to 30 µL. Amplification step: pre denaturation at 95 °C for 5 min, 95 °C denaturation for 30 s, 55 °C annealing for 30 s, 72 °C extension for 45 s, 27 cycles, 72 °C extension for 10 min. The amplified product was detected with 1% agarose gel and sent to Sangon (Shanghai, China) for sequencing based on Illumina Miseq platform. The original sequencing sequence was subjected to quality control and processing analysis, and the differences in microbial diversity and microbial community structure of rumen fluid and feces of fattening sheep in CON and MHB groups were analyzed. 

### 2.4. Metabolomics Detection of Rumen Fluid, Serum and Urine

Ultra-high-performance liquid chromatography-tandem quadrupole time-of-flight mass spectrometry (UPLC-Q-TOF-MS) was used to analyze the metabolomics of rumen fluid, serum, and urine samples stored at −80 °C in the early stage of fattening sheep. The flow rate of the test sample was 0.5 mL/min, and the column temperature was 25 °C, the mobile phase consists of solvent A (an aqueous solution of 25 mM ammonium acetate and 25 mM ammonia water, pH = 9.75) and solvent B (acetonitrile). The ionization source of LC-QTOFMS platform is electric spray ionization, including positive ion mode (POS) and negative ion mode (NEG). The combination of two methods for detecting metabolites has higher coverage. After quality control of the raw data obtained from UPLC-Q-TOF-MS, multivariate statistical analysis was conducted. The *p*-value of the Student’s *t*-test was less than 0.05, and the VIP value of the Variable Importance in the Projection (VIP) of the OPLS-DA model’s first principal component was greater than 1. Differential metabolites were screened, and the results were visualized in volcanic map form. The hierarchical cluster analysis of differential metabolites was conducted, the metabolic pathway analysis of differential metabolites was conducted to analyze the differences in metabolites and metabolic pathways between CON group lambs and MHB group lambs.

### 2.5. Transcriptomics Analysis

Transcriptomic analysis was performed on sheep rumen epithelial tissue using Illumina Hiseq™, The original image data file was obtained and transformed into the original sequenced reads through CASAVA base calling analysis. FastQC was used to visually evaluate the quality of the sample sequencing data, and cutadapt was used to remove joints, trimmatic to remove low-quality bases at both ends and reads filtering were performed. Data quality control was also carried out. Analysis was conducted using DESeq2, with screening conditions set to *p*-value < 0.05 and multiple differences |FoldChange| > 2. The results of differential analysis were presented in the form of a volcanic map. Using GO functional enrichment to analyze the pathways of differentially expressed genes (DEGs), the biological functions involved in differentially expressed genes (DEGs) were explored, and the molecular functions, cellular components, and biological processes involved were described. ClusterProfiler was used for functional enrichment analysis, and when *p*-value < 0.05, it is considered that the function is enriched.

### 2.6. Statistical Analysis

The SPSSAU data analysis platform (https://spssau.com/ accessed on 6 October 2023) was used for the normal distribution test of the experimental data. The data statistical analysis was conducted using SPSS software (version 20.0, Chicago, IL, USA) for independent sample *t*-tests. When *p* < 0.05 is a significant difference, and *p* < 0.01 is a highly significant difference, there were 6 replicates of the experimental samples.

## 3. Results

### 3.1. Growth and Performance

Before the experiment, the body weight of fattening sheep in the CON and MHB groups was measured, and the initial average body weight difference was not significant (MHB group 23.35 ± 2.06 kg vs. CON group 23.25 ± 1.54 kg). After two months of feeding, the MHB and CON groups of sheep gained 13.43 kg and 11.19 kg, respectively (MHB group 36.78 ± 2.27 kg vs. CON group 34.45 ± 1.34 kg). The average daily feed intake of MHB group sheep increased by 3.03% compared to the CON group, and the average daily weight gain of MHB group sheep increased by 20.1% compared to the CON group, The feed conversion efficiency of the MHB group was 2.49% higher than that of the CON group, and feeding a diet containing 8% MHB as a substitute significantly improved the growth performance of fattening sheep.

### 3.2. Characteristics of Intestinal Tissue Morphology

Due to the high content of crude fiber in the diet of meat sheep, which is fermented by rumen microorganisms and digested and absorbed by the intestine, we determined the effect of MHB diet on the intestinal digestion and absorption function of fattening sheep by observing the morphology of various intestinal tissues. We observed the duodenum, jejunum, and ileum of the small intestine, as well as the cecum, colon, and rectum of the large intestine, and measured the thickness of the intestinal mucosal muscle layer, villus length, mucosal muscle layer, and mucosal layer. The morphological structure of various intestinal segments of meat sheep was observed under an optical microscope as shown in the figure (Figure 2), and the statistical data of the measurement results are shown in the table (Table 1). Compared with the CON group, the MHB group showed a significant increase in the length of ileal villi (*p* < 0.01), a significant decrease in the thickness of the cecal mucosal layer (*p* < 0.01), and a significant increase in the thickness of the colonic mucosal layer and rectal mucosal muscle layer (*p* < 0.05).

### 3.3. Analysis of Rumen Fermentation Parameters and Rumen Microbiota

As shown in Table 2, the ammonia nitrogen levels in the rumen fluid of the MHB group were lower than those of the CON group, with no significant difference. The concentration of total volatile fatty acids (VFA) was significantly higher than that of the CON group (*p* < 0.05), and the concentration of acetic acid was significantly higher than that of the CON group (*p* < 0.05). There was no significant difference in the concentration of propionic acid, butyric acid, and valeric acid in the feed and the ratio of acetic acid to propionic acid between the two groups (*p* > 0.05).

The classification of rumen microbiota at the genus level was shown in the figure (Figure 3A), with a total of 260 taxa identified (Appendix A). Among them, Prevotella (28.19 ± 4.93%), Methanobrevibacter (5.86 ± 3.51%), and Succiniclassicum (4.41 ± 3.04%) were the three dominant genera in the rumen of meat sheep. The main microbial communities with significant differences between the CON and MHB groups (accounting for 0.05% of the total sequence in at least one sample) were shown in the table (Table 3), among which Paraprevotella, Alloprevotella, Marinilabilia, Saccharibacteria_general_incertae_sedis, Blautia, Subdivision5_general_incertae_sedis and Ornatilina have a significant difference in 7 genera between the CON group and the MHB group, with Paraprevotella, Alloprevotella, Marinilabilia, Saccharibacteria in the MHB group_general_incertae_sedis, Subdivision5_general_incertae_. The relative abundance of Sedis and Ornatilina was significantly higher than that of the CON group (*p* < 0.05), while the proportion of Blautia in the MHB group was significantly lower than that in the CON group (*p* < 0.05).

### 3.4. Analysis of Fecal Microbiota

The distribution of fecal microbiota at the genus level was shown in the figure (Figure 3B), with a total of 230 taxa identified (Appendix A). Among them, Sporobacter (8.90 ± 2.60%), Bacteroides (7.01 ± 2.28%), and Treponema (4.20 ± 3.00%) were the three dominant genera in sheep feces. The main fecal microbiota with significant differences between the CON and MHB groups were shown in Table 4. Among them, five genera, Prevotella, Blautia, Coprococcus, Clostridium XlVb, and Parasutterella, showed significant differences between the CON and MHB groups. The proportion of Prevotella, Clostridium XlVb, and Parasutterella in the MHB group was significantly higher than that in the CON group (*p* < 0.05), while the proportion of Blautia and Coprococcus in the MHB group was significantly lower than that in the CON group (*p* < 0.05).

### 3.5. Metabolomics Analysis of Rumen Fluid

We used a non-targeted LC-MS method to evaluate metabolites in the rumen fluid samples of 12 fattening sheep in the MHB and CON groups. The standard was set to a *p*-value of less than 0.05 in the Student’s *t*-test, and the variable projection importance (VIP) of the first principal component of the OPLS-DA model was greater than 1. There were 105 metabolites screened for significant differences between the MHB and CON groups, of which 34 were negative modes and 71 were positive modes (Appendix A).

The rumen metabolites of the MHB group were aggregated separately from those of the CON group (Figure 4). Compared with the CON group, the MHB group showed significant downregulation of 14 differential metabolites, including Ala-Ala, Daidzein, and Methyldopa (*p* < 0.05), while 91 differential metabolites such as Phenol, Salicylic acid and Phenylacetic acid were significantly upregulated (*p* < 0.05). The significantly altered metabolites between the MHB group and the CON group produced 15 metabolic pathways (Appendix A). We found that metabolic pathways, such as pyrimidine metabolism, taurine and taurine metabolism, and nicotinate and nicotinamide metabolism, had a significant impact (Figure 5).

### 3.6. Serum Antioxidant Capacity and Metabolomics Analysis

Firstly, we evaluated the effect of mint on the antioxidant capacity of sheep serum. In Table 5, it can be seen that compared with the CON group, the serum GSH-Px activity (MHB group 351.29 ± 50.31 U/mL vs. CON group 287.42 ± 35.36 U/mL, *p* < 0.05), SOD activity (MHB group 20.06 ± 0.53 U/mL vs. CON group 15.55 ± 1.67 U/mL, *p* < 0.05), and T-AOC level (MHB group 0.56 ± 0.19 U/mL vs. CON group 0.29 ± 0.12 U/mL, *p* < 0.05) in the MHB group increased by 22.22%, 30.29%, and 93.10%, respectively. The difference in MDA content (MHB group 2.09 ± 1.24 nmol/mL vs. CON group 2.97 ± 1.07 nmol/mL, *p* > 0.05) was not significant, indicating that MHB has the ability to improve the antioxidant capacity of mutton sheep serum.

Secondly, we screened a total of 163 serum metabolic markers between the MHB group and the CON group. Among them, there were 53 anionic patterns and 110 cationic patterns that showed significant changes (Appendix A). Cluster analysis was performed on the differential metabolites obtained, compared with the CON group, the MHB group showed significant changes: 37 differential metabolites, such as Citric acid, Confertifoline and Sedoheptolose were significantly upregulated; 126 differential metabolites, such as Sepiapatin, Salicylate, and N-Acetylneuraminic acid, were significantly downregulated (Figure 6). The significantly altered metabolites between the MHB and CON groups produced 30 metabolic pathways (Appendix A). The results of metabolic pathway analysis were presented in bubble plots, it can be seen that metabolic pathways such as glycerol ester metabolism, pentose phosphate pathway phosphate pathway, and glycine, serine, and threonine metabolism have a significant impact (Figure 7). 

### 3.7. Urine Metabolomics Analysis

A total of 54 urine metabolic markers were screened between the MHB and CON groups, with 24 anionic patterns and 30 cationic patterns showing significant changes (Appendix A). Compared with the CON group, the MHB group showed significant downregulation of 19 differential metabolites, including Phosphocreatine, 4-Hydroxybenzaldehyde, and Cysteine glycine (*p* < 0.05), and a total of 35 significantly upregulated metabolites, including 5′-Deoxyadenosine, Acetylhexamide, and 1-Naphthol (*p* < 0.05) (Figure 8). Metabolic pathway analysis revealed that metabolites with significant changes between the MHB and CON groups produced 15 metabolic pathways (Appendix A). Glutathione metabolism, valine, leucine, and isoleucine biosynthesis, terpenoid backbone biosynthesis and glyoxylate and dicarboxylate metabolism had a significant impact (Figure 9).

### 3.8. Analysis of Circular RNA in Rumen Epithelial Tissue

We screened a total of 14 circRNA expression differences between the MHB group and the CON group, with nine upregulated and five downregulated (Appendix A and Figure 10A). GO analysis was used to investigate the distribution of differentially expressed circRNAs in annotation functions. Differentially expressed circRNA host genes were mainly enriched in cellular processes, metabolic processes, biological regulation, and regulation of biological processes. The cell locations were mainly cells, cell components, organelles, and organelle parts. The molecular function mainly focused on catalytic activity and binding function (Figure 10B). We conducted functional enrichment analysis, and the top 30 differentially expressed circRNA genes with the highest degree of significant functional enrichment were screened (Appendix A and Figure 10C). The differentially expressed circRNA gene function is mainly significantly enriched in biological processes related to improving immune function and histone modification.

### 3.9. Analysis of lncRNA in Rumen Epithelial Tissues

The results of significant differences in lncRNA and mRNA expression between the MHB group and the CON group were screened (Appendix A). Among them, four lncRNAs were significantly differentially expressed, with one being significantly upregulated and three being significantly downregulated. There were 23 mRNA differentially expressed, nine significantly upregulated, and 14 significantly downregulated (Figure 11A). The differentially transcribed target genes were mainly enriched in cellular processes, metabolic processes, and biological regulation in biological processes. The proportion of cell and cellular categories in cell component classification was high, while the proportion of catalytic activity and binding in molecular functions was high (Figure 11B), The significant enrichment of gene functions corresponding to differentially expressed transcripts mainly occurs in biological processes such as cell migration, which is a physiological process of normal development of the body. Processes such as angiogenesis, wound healing, immune response, and inflammatory response all involve cell migration, which is consistent with the efficacy of MHB (Appendix A and Figure 11C).

### 3.10. miRNA Analysis of Rumen Epithelial Tissue

Significant differences in miRNAs between the MHB and CON groups were identified through screening (Appendix A), with 217 significantly upregulated and 131 significantly downregulated (Figure 12A). GO enrichment analysis was conducted using targeted mRNA genes with significantly differentially expressed miRNAs as the target genes. Among them, cellular processes, single organism processes, and biological regulation were the biological processes with the most enriched genes. The cellular components are mainly enriched in cells, organelles, and membranes. The molecular functions are mainly catalytic activity and binding function (Figure 12B). At the same time, we selected the top 10 functional analysis results with the highest enrichment in gene molecular function, cellular location, and biological processes involved. The differentially expressed miRNAs correspond to gene functions that are mainly enriched in biological processes such as transcriptional regulation and protein phosphorylation (Appendix A and Figure 12C), which are of great significance for gene expression regulation, regulating protein activity and function, as well as regulating cell metabolism and proliferation. This indicates that adding MHB to the diet has a positive effect on fattening sheep breeding.

## 4. Discussion

### 4.1. Growth Performance and Intestinal Tissue Morphology

Recent studies have shown that MHB has strong anti-inflammatory, analgesic, sedative, and appetizing functions. Adding an appropriate amount of MHB to feed can not only improve the taste of the feed, promote digestion and absorption of the feed, and improve feed utilization, but also enhance animal immunity and antioxidant capacity [12,13]. In the breeding of laying hens, MHB can improve egg production and nutritional value, especially in preventing and treating diseases such as indigestion and intestinal inflammation. It can also improve the growth performance and rumen fluid physicochemical indicators of beef cattle [13,14,15]. In this study, after 56 days of feeding, the total body weight of the MHB group fattening sheep was higher than that of the CON group, and the average daily feed intake, average daily weight gain, and feed conversion rate of the MHB group were all improved, which is consistent with the research results of adding MHB to the diet of broilers [12]. This may be partly due to the unique aromatic aroma of MHB and its presence in menthol, which can improve the digestion and absorption of dietary nutrients and is an appetizer that helps promote animal feeding in animal production [12,16]. On the other hand, MHB has functions, such as dispersing wind, dissipating heat, soothing liver, detoxifying, and promoting liver protection. MHB essential oil has broad-spectrum antibacterial properties, which can inhibit microbial growth and promote animal health, indirectly promoting animal production performance [10,17,18].

The length of small intestinal villi and the thickness of the muscle layer are morphological indicators that measure the duodenum and jejunum. There are no villi in the cecum and colon, and the intestinal contractility of the large intestine can be determined by the thickness of the mucosal layer and mucosal muscle layer. The results showed that the thickness of the duodenal muscle layer and the length of the jejunal villi in the MHB group were higher than those in the CON group, but the difference was not significant. The length of the ileum villi was significantly higher than that in the CON group, while the colonic and rectal mucosal muscle layers were significantly higher than those in the CON group. The cecal mucosal layer was significantly lower than that in the CON group, and the length of the jejunal villi in the MHB group increased by 30.93% compared to the CON group, indicating a stronger trend in the digestion and absorption ability of the jejunum in the experimental group. The thicker duodenal muscle layer indicates a higher dietary fiber level in the MHB group, proving that MHB has a promoting effect on the intestinal digestion function of meat sheep. A high-fiber diet can cause changes in the morphology of the cecum [19], while a high-protein diet can cause changes in the colon structure and mucosal morphology [20]. 

The cecum is located at the beginning of the large intestine segment, which can control the flow rate of small intestine content into the large intestine segment and prevent the reflux of large intestine content into the small intestine segment. The thickness of the mucosal layer of the cecum in the MHB group is significantly reduced and the mucosal muscle layer is lower than that in the CON group. This indicates that the contraction ability of the cecum in the MHB group is weak, which can prolong the retention time of food in the small intestine and promote the absorption of nutrients by the body. The thickness of the colonic and rectal mucosal muscle layers increases, indicating that the MHB group has increased colonic and rectal peristalsis, which is beneficial for absorbing small amounts of water, glucose, and other substances in the feces. This is also beneficial for the excretion of feces, reducing the burden on the gastrointestinal tract, and promoting gastrointestinal peristalsis to better digest and absorb nutrients in the feed. This indicates that feeding MHB has a promoting effect on the absorption of intestinal nutrients and the elimination of metabolic waste.

### 4.2. Rumen and Fecal Microorganisms

In this study, supplementing MHB significantly increased the concentration of VFA, indicating that MHB has a positive effect on rumen microbial fermentation. The level of VFA is positively correlated with the energy supply of fattening sheep. Therefore, the average daily weight gain and feed utilization rate of the MHB group were improved, and the abundance of seven microbial species showed significant changes at the genus level. Paraprevotella, Alloprevotella, and others in the MHB group were higher than those in the CON group, Paraprevotella degrades carbohydrates and polysaccharides in food and participates in vitamin synthesis in the body [21]. Alloprevotella, as a beneficial bacterium, can improve the intestinal barrier and exert anti-inflammatory effects [22]. The rumen, as an important site for microorganisms to ferment dietary nutrients into VFAs, ammonia, and other nutrients [23], has a significant impact on the physiological function and physical health of ruminants. The addition of MHB to the diet improves the production performance of fattening sheep by adjusting the rumen microbiota. At the genus level, the abundance of five types of microorganisms added to the diet has changed, Prevotella plays a crucial role in fiber degradation [24], and Parasutterella is a core member of the gut microbiota, which may be involved in maintaining bile acid homeostasis and cholesterol metabolism [25]. It can be seen that the upregulated microbiota in the MHB group is positively correlated with average daily weight gain, feed conversion rate, etc. Downregulation of the relative abundance of bacterial genera Blautia and Coprococcus contributes to weight loss, and research has found that a large number of phenotypes exist in lean individuals [26]. This indicates that adding MHB to the diet helps promote the proliferation of beneficial gut microbiota, thereby improving the growth performance of fattening sheep and the feed reward ratio.

### 4.3. Metabolomics of Rumen Fluid, Serum, and Urine

We conducted metabolomics analyses on the rumen fluid, serum, and urine of MHB and CON groups of fattening sheep to further understand how mint can improve the growth performance of fattening sheep. Metabolomics analysis found that most metabolites in the rumen fluid, serum, and urine have high levels, mainly including amino acids, dipeptides, fatty acids, and carbohydrates. The significant changes in rumen fluid are mainly pyrimidine metabolism, taurine and taurine metabolism. The main metabolic pathways in the serum are glyceride metabolism, the pentose phosphate pathway, and glycine, serine, and threonine metabolism. The main metabolic pathways in the urine are glutathione metabolism, valine, and biosynthesis of leucine and isoleucine; biosynthesis of terpenoid skeletons; and metabolism of glyoxylic acid and dicarboxylic acid. The pyrimidine metabolism pathway can achieve the recycling and reuse of pyrimidine within cells and can also provide nitrogen or carbon sources for microbial growth [27]. Taurine and taurine metabolism can enhance the digestion and absorption of lipids in the gastrointestinal tract and exhibit anti-inflammatory and antioxidant effects in diseases that affect animal production, such as animal heat stress, gastrointestinal injury, and mastitis [28]. Glycerol ester metabolism is a type of fat metabolism involved in regulating energy conversion, material transport, cell differentiation, and the development of cells [29]. Glutathione metabolism plays an important role in organisms, including anti-infection, antioxidant function, and participation in substance absorption in the immune system [30]. Based on metabolomics analysis of rumen fluid, serum, and urine, adding MHB to the diet of fattening sheep can affect the levels of metabolites in rumen fluid, serum, and urine, leading to changes in related metabolic pathways. These differential metabolites and metabolic pathways are often related to energy metabolism, fatty acid synthesis, disease occurrence, and the immune ability of the fattening sheep body, which can improve the antibacterial and antioxidant levels of fattening sheep, indirectly improving the growth performance of fattening sheep.

### 4.4. Rumen Epithelial Tissue Transcriptomics

CircRNA is a special class of non-coding RNA molecules that exhibit a closed circular structure, are not affected by RNA exonucleases, are more stable in expression, are less prone to degradation, and play an important role in organisms, with significant research value [31,32,33]. LncRNA is a type of non-coding RNA with a length greater than 200 nucleotides, which plays an important role in numerous life activities and is a hot topic in genetic research [34]. MiRNAs are a class of non-coding single stranded RNA molecules with a length of approximately 22 nucleotides encoded by endogenous genes, involved in post transcriptional gene expression regulation in animals and plants [35,36]. We performed transcriptome sequencing and differentially expressed genes (DEGs) analysis on rumen epithelial tissue, and circRNA sequencing results showed that a total of nine circRNAs were significantly upregulated and five were significantly downregulated between the MHB and CON groups. LncRNA sequencing analysis showed a significant upregulation of 10 and a significant downregulation of 17. miRNA sequencing analysis showed a significant upregulation of 217 and a significant downregulation of 131. GO enrichment analysis of differentially expressed circRNA, lncRNA, and miRNA revealed that differentially expressed circRNA mainly involve functions related to immune response, energy metabolism, and protein synthesis. Further functional enrichment analysis showed that the most enriched functions mainly include protein modification, inflammatory response, and oxidative stress response. This is consistent with the anti-inflammatory, antioxidant, and digestive promoting effects of MHB, confirming that feeding a diet containing MHB promotes the anti-inflammatory, antioxidant, and other abilities of the body of fattening sheep, which is of great significance for the healthy growth and improvement of their growth performance of fattening sheep.

According to the reports, peppermint essential oil (PEO) contained in MHB can improve gastric and intestinal motility, reduce satiety and functional dyspepsia, and promote digestion and absorption [37,38]. In addition, PEO can increase liver glutathione levels, improve liver function and antioxidant activity [39]. In the agricultural field, PEO is added to animal diets to exert antibacterial and anti-inflammatory effects [40]. These studies are consistent with the results of this experiment, verifying the anti-inflammatory and antibacterial effects of adding an appropriate amount of mint to the diet of fattening sheep, reducing inflammation and stress reactions in the body, regulating gastrointestinal microbiota, thereby improving the health status of fattening sheep, and indirectly improving their growth performance.

MHB, as a perennial herbaceous plant, is also a common traditional Chinese medicine resource. Replacing 8% peppermint with 8% feed can reduce the use of antibiotics, promote healthy breeding, provide high-quality and safe lamb products for humans and reduce the cost of fattening sheep, resulting in higher breeding economic benefits. In the future, in-depth research will be conducted on the molecular pathways of MHB in vivo, explaining the specific mechanisms of MHB in healthy breeding processes, and providing an important theoretical basis for the widespread application of MHB in the field of animal husbandry.

## 5. Conclusions

In summary, adding mint to the diet increased the thickness of duodenal and jejunal villi, significantly increased the thickness of ileal villi, colonic mucosa, and rectal mucosa, significantly reduced the thickness of cecal mucosa, improved intestinal digestion function, improved rumen fermentation, changed rumen bacterial composition, and increased Paraprevotella, Alloprevotella, Marinilabilia, Saccharibacteria_general_incertae_sedis, Subdivision5_general_incertae_. The relative abundance of Sedis and Ornatilina decreased the abundance of Blautia while effectively improving the gut microbiota, increasing the relative abundance of beneficial gut bacteria Prevotella, Clostridium XlVb, and Parasutterella, and reducing the relative abundance of Blautia and Coprococcus. In the MHB group, the increased levels of metabolites in rumen fluid, serum, and urine were mainly used as substrates for microbial protein synthesis. This indicates that supplementing MHB in the diet to improve production performance is mainly related to protein synthesis, and multiple metabolic pathways are closely related to protein synthesis and energy metabolism, which are of great significance for the growth performance and body health of fattening sheep, including pyrimidine metabolism, taurine and taurine metabolism, glycerol ester metabolism, the pentose phosphate pathway and glutathione metabolism, as well as transcriptome sequencing analysis of rumen epithelial tissue, revealed that differentially expressed genes mainly focus on functions related to immune response, energy metabolism, and protein modification, thereby improving the health level and growth performance of fattening sheep. The experimental results not only provide insights for the role of MHB as a substitute for antibiotics in ruminant feed but also provide insights for the supplementation of appropriate MHB in the diet to improve the fattening effect of fattening sheep.

## Figures and Tables

**Figure 1 animals-13-03461-f001:**
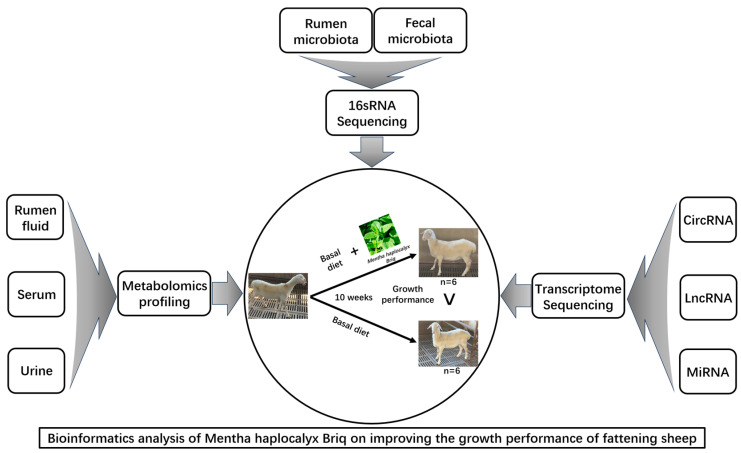
Experimental design and workflow of feeding fattening sheep with mint diet. Including rumen and fecal microbiome, rumen fluid, serum and urine metabolome, and transcriptome sequencing of rumen epithelial cells. Twelve Hu sheep were randomly assigned to a basal diet (CON), or a basal diet supplemented with 80 g/kg peppermint DM (MHB).

**Figure 2 animals-13-03461-f002:**
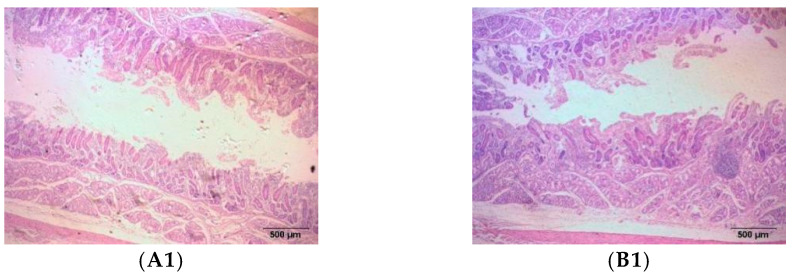
Histomorphology of intestinal segments in fattening sheep fed with MHB diet. (**A**) CON, (**B**) MHB; (**1**) Duodenum, (**2**) Jejunum, (**3**) Ileum, (**4**) Cecum, (**5**) Colon, (**6**) Rectum.

**Figure 3 animals-13-03461-f003:**
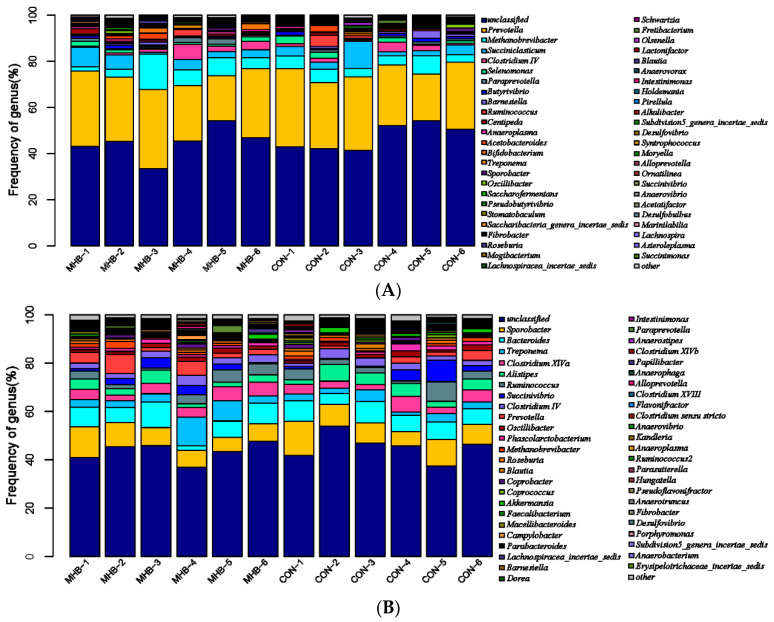
Distribution characteristics of horizontal microbiota in the MHB group. (**A**) Distribution of rumen microbiota, (**B**) distribution of fecal microbiota.

**Figure 4 animals-13-03461-f004:**
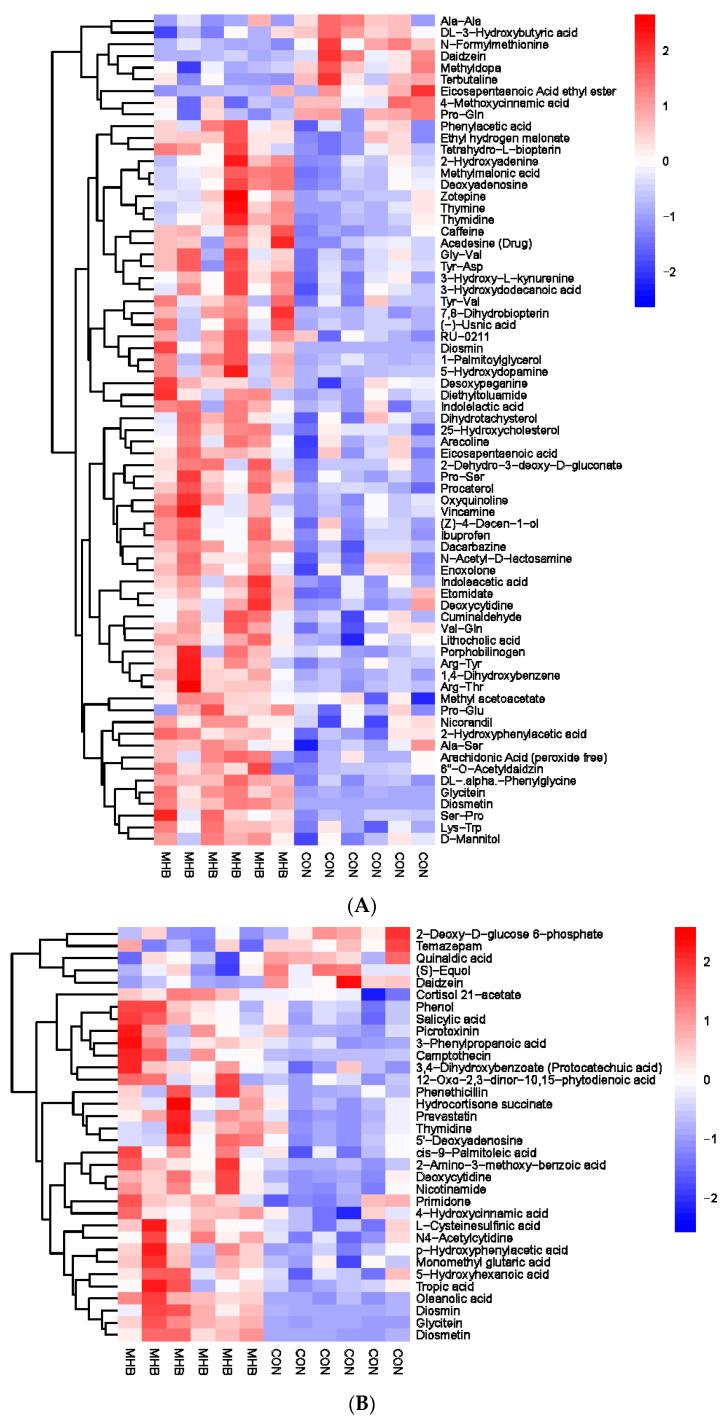
Hierarchical cluster analysis of rumen metabolites in fattening sheep. (**A**) Hierarchical clustering analysis of MHB groups in cationic mode, (**B**) hierarchical cluster analysis of MHB Group in anionic mode. Note: The horizontal axis represents different experimental groups, the vertical axis represents the differential metabolites compared in the group, the blue color block represents the relative expression level of the corresponding position metabolites is down regulated, and the red color block represents the up regulation.

**Figure 5 animals-13-03461-f005:**
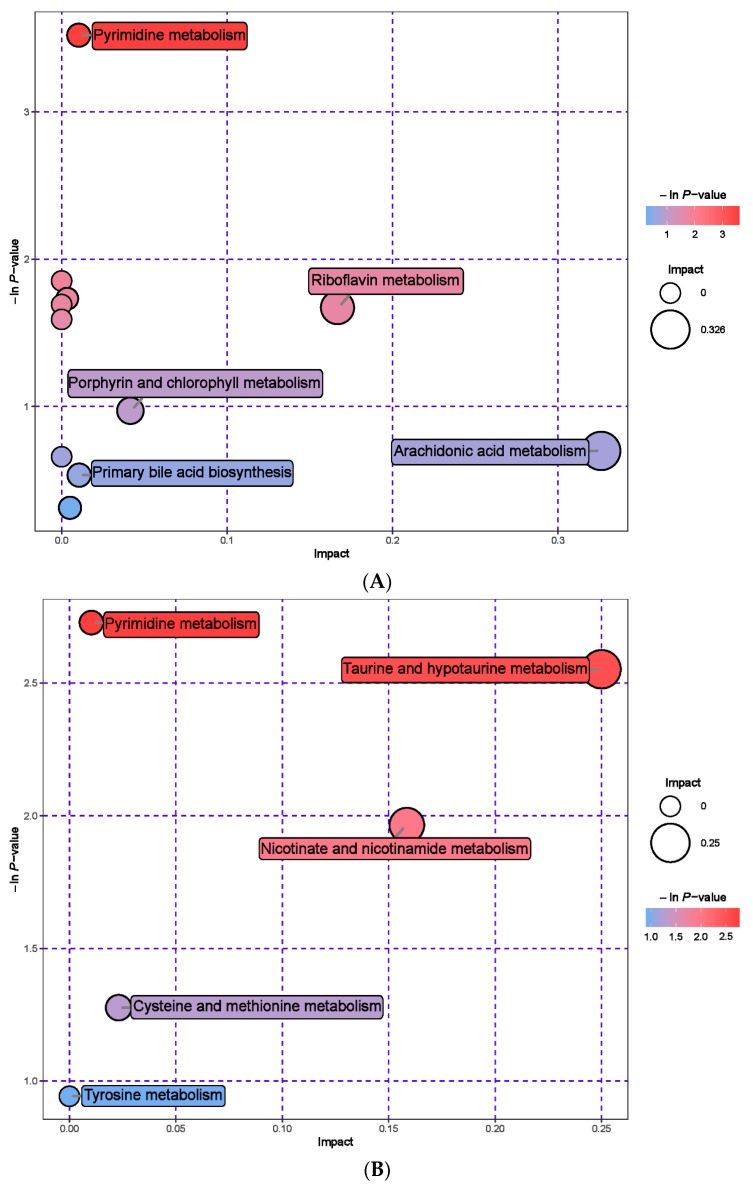
Pathway analysis of rumen metabolomics in fattening sheep. (**A**) Pathway analysis of MHB group in cationic mode, (**B**) pathway analysis of MHB group in anionic mode. Note: The color and size of bubbles indicate the impact of mint treatment on sample metabolism, while larger red bubbles indicate a greater impact on the pathway.

**Figure 6 animals-13-03461-f006:**
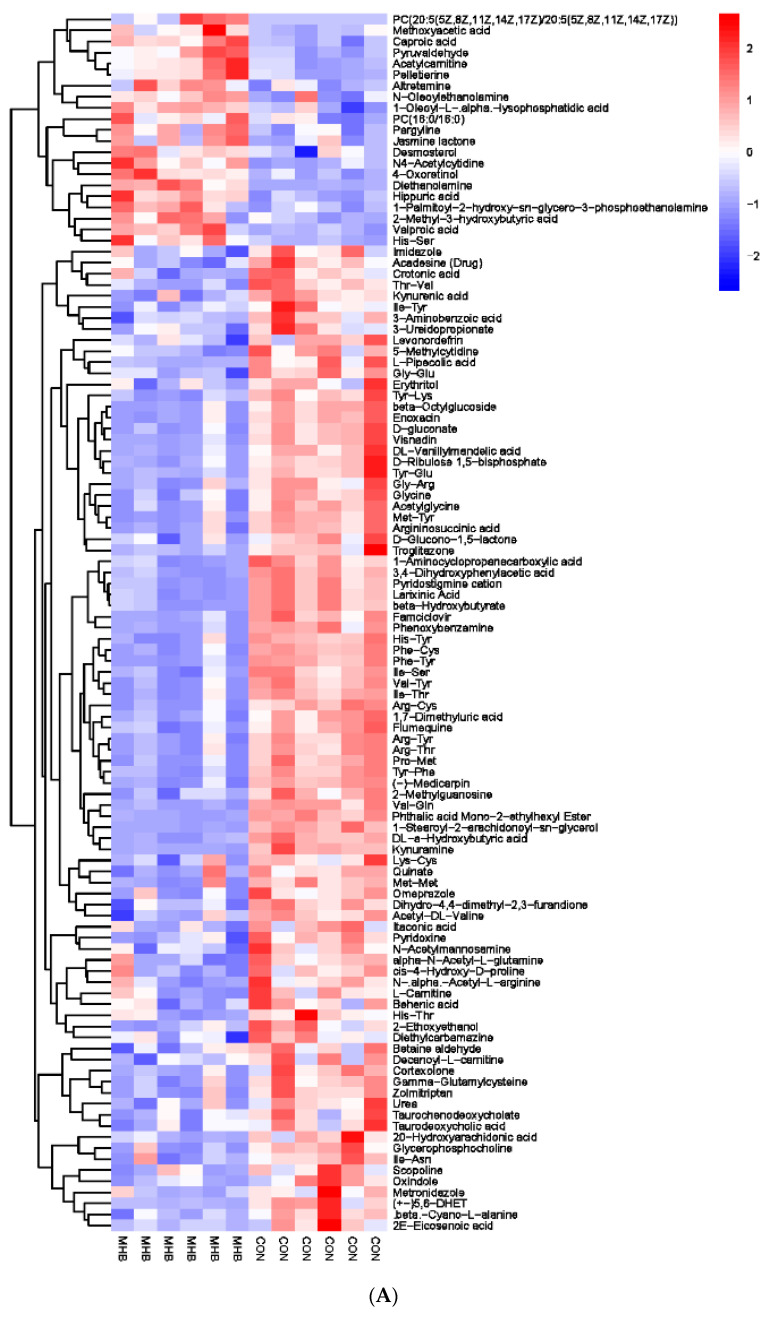
Hierarchical cluster analysis of serum metabolites in fattening sheep. (**A**) Hierarchical cluster analysis of MHB group serum in cationic mode, (**B**) hierarchical cluster analysis of MHB Group serum in anionic mode.

**Figure 7 animals-13-03461-f007:**
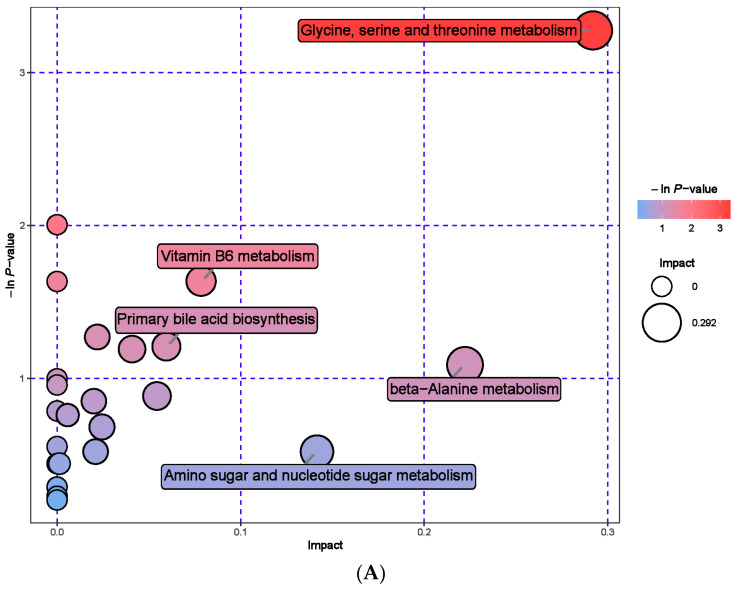
Pathway analysis of serum metabolomics. (**A**) Pathway analysis of MHB group serum in cationic mode, (**B**) pathway analysis of MHB group serum under anionic mode.

**Figure 8 animals-13-03461-f008:**
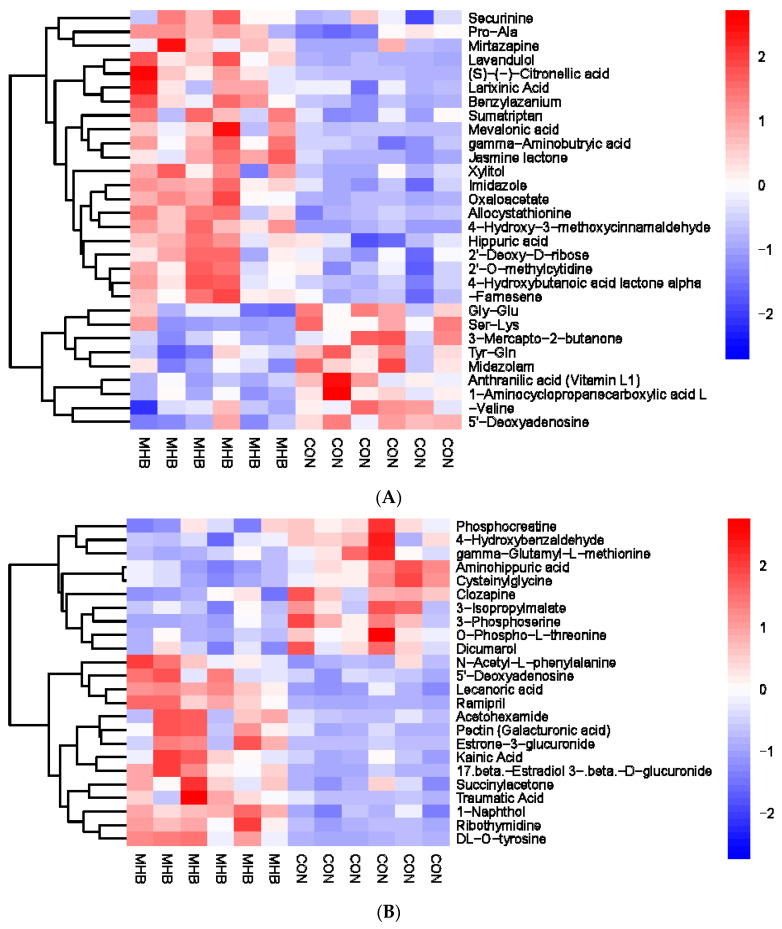
Hierarchical cluster analysis of urine metabolites in fattening sheep. (**A**) Hierarchical cluster analysis of MHB group urine in cationic mode, (**B**) hierarchical cluster analysis of MHB group urine in anionic mode.

**Figure 9 animals-13-03461-f009:**
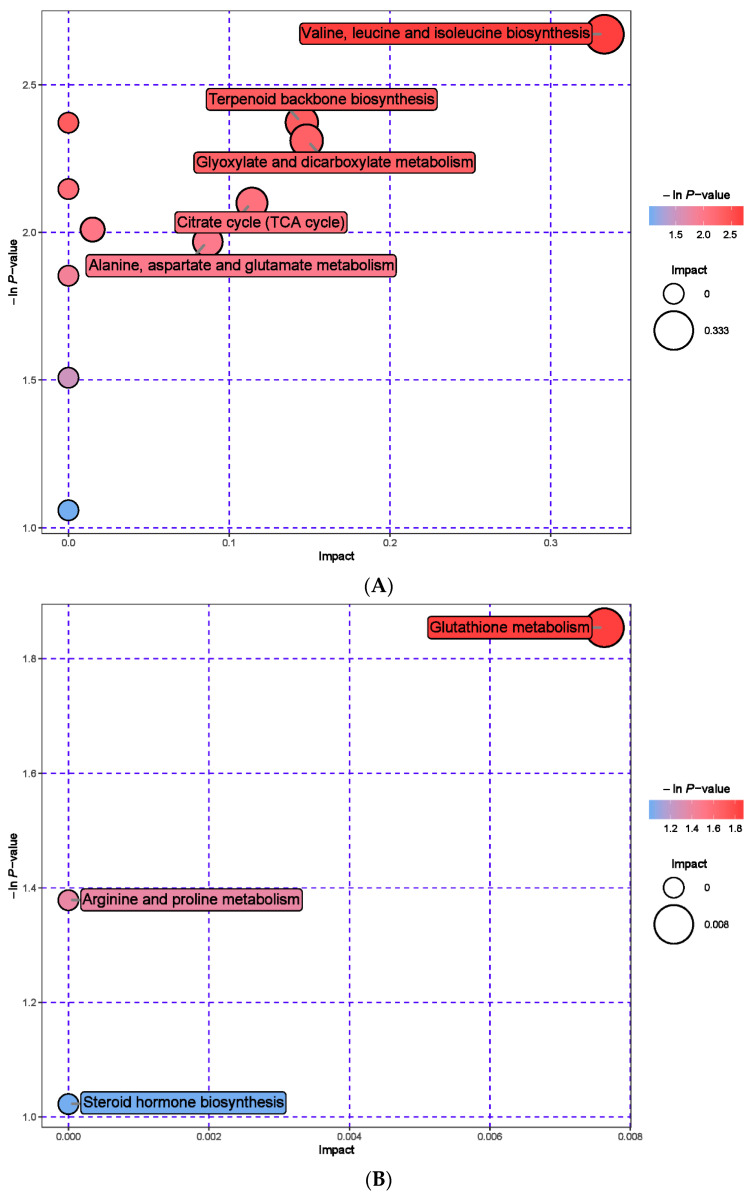
Pathway analysis of urine metabolomics. (**A**) Pathway analysis of MHB group urine in cationic mode, (**B**) pathway analysis of MHB group urine under anionic mode.

**Figure 10 animals-13-03461-f010:**
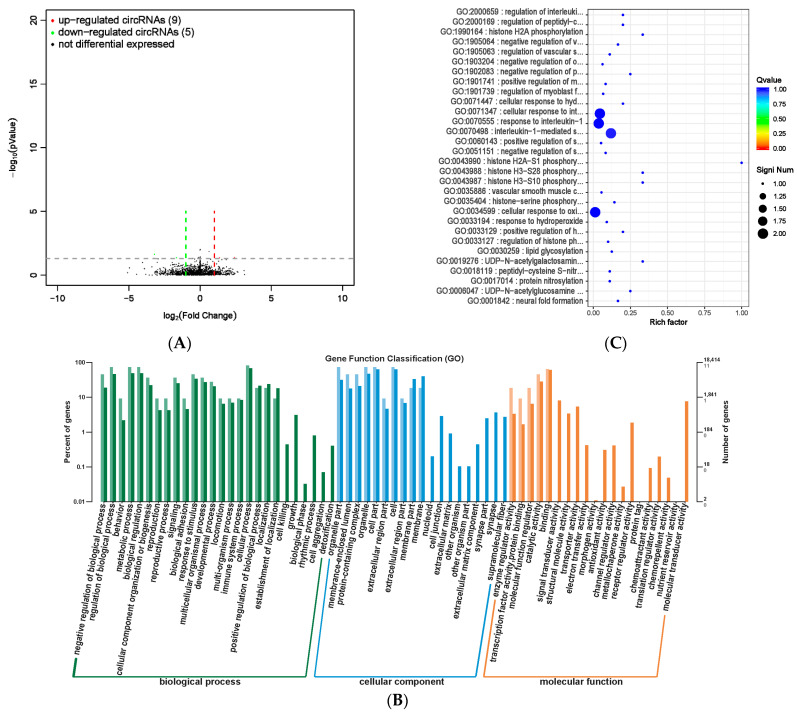
Differential expression analysis of circRNA between MHB group and CON group. (**A**) Volcano map of circRNA expression differences between MHB group and CON group. The horizontal axis represents the fold change (log (B/A)) value of the transcript expression difference between different groups, while the vertical axis represents the *p*-value of the transcript expression change. The smaller the *p*-value, the greater the −log (*p*-value), and the more significant the difference. Red represents upregulated transcripts, green represents downregulated transcripts, and black represents non differential transcripts, (**B**) histogram of host gene functional annotation classification for differentially expressed circRNA between MHB and CON groups. The horizontal axis represents the functional classification, while the vertical axis represents the number of genes within the classification (**right**) and their percentage in the total number of annotated genes (**left**). Light colors represent host genes, while dark colors represent all genes, (**C**) the top 30 functional scatter plots show significant enrichment of circRNA between the MHB group and the CON group. The vertical axis represents functional annotation information, while the horizontal axis represents the Rich factor corresponding to the function. The size of the Qvalue is represented by the color of the dot. The smaller the Qvalue, the closer the color is to red. The number of differentially expressed circRNA host genes is represented by the size of the dot.

**Figure 11 animals-13-03461-f011:**
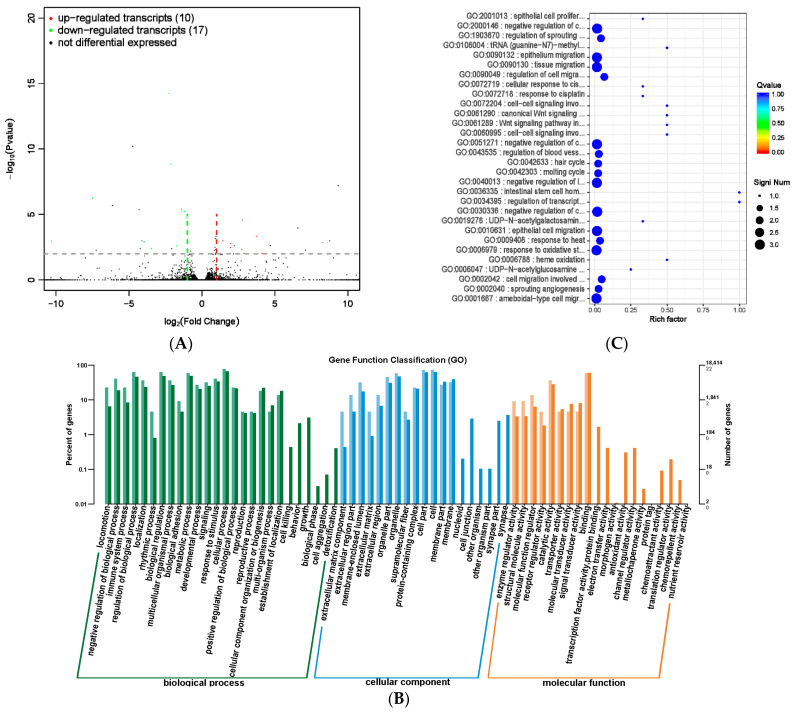
Differential expression analysis of transcriptome between MHB group and CON group. (**A**) Volcano map of transcript expression differences between MHB group and CON group, (**B**) histogram of functional annotation classification of genes corresponding to MHB and CON differential transcripts, (**C**) the top 30 functional scatter plots show significant enrichment of differentially expressed transcripts in the MHB and CON groups.

**Figure 12 animals-13-03461-f012:**
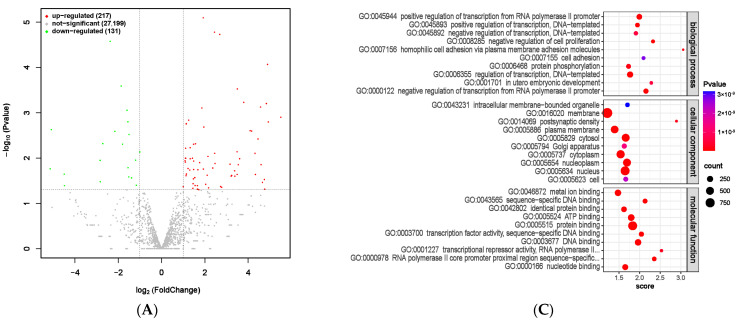
Differential expression analysis of miRNA between MHB group and CON group. (**A**) Volcano map of miRNA expression differences between MHB group and CON group. (**B**) MHB group and CON group differentially expressed miRNA GO annotation classification histogram. (**C**) The functional scatter plot shows a significant difference in miRNA enrichment between the MHB group and the CON group, with a top 10 × 3 degree (biological process, cellular component, and molecular function each with a top 10 degree).

**Table 1 animals-13-03461-t001:** Effect of MHB on the Morphology of Intestinal Segment in Fattening Sheep.

Position	Item	CON (μm)	MHB (μm)
Duodenum	Muscularis	431.17 ± 3.29	445.01 ± 39.53
	Villus	383.11 ± 5.00	334.62 ± 29.51
Jejunum	Muscularis	272.98 ± 5.24	219.98 ± 35.05
	Villus	362.08 ± 38.72	474.07 ± 16.56
Ileum	Muscularis	161.41 ± 7.32	164.53 ± 12.63
	Villus	276.27 ± 4.31	201.45 ± 9.86 **
Cecum	Muscularis mucosae	344.66 ± 11.46	338.46 ± 13.99
	Mucosa	396.35 ± 14.37	246.23 ± 8.03 **
Colon	Muscularis mucosae	276.68 ± 12.00	244.32 ± 7.16
	Mucosa	535.91 ± 23.65	627.06 ± 16.09 *
Rectum	Muscularis mucosae	363.97 ± 8.92	418.78 ± 11.66 *
	Mucosa	351.72 ± 32.90	423.86 ± 9.76

Note: * indicates a significant difference (*p* < 0.05). ** indicates a highly significant difference (*p* < 0.01).

**Table 2 animals-13-03461-t002:** Rumen fermentation characteristics of peppermint diet treatment.

Item	Treatment		SEM	*p*-Value
	CON	MHB		
Ammonia-nitrogen, mg/dL	6.21	4.00	1.1119	0.065
Total volatile fatty acid, mM	66.10	73.15	2.9304	0.029
Acetate, mM	45.23	51.2	2.4436	0.027
Propionate, mM	12.21	13.4	0.9084	0.193
Butyrate, mM	7.85	7.72	0.5670	0.820
Valerate, mM	0.81	0.78	0.0456	0.592
Acetate: Propionate	3.71	3.94	0.3498	0.530

**Table 3 animals-13-03461-t003:** Significant changes in the main microbial communities in the rumen of the MHB diet group.

Phylum	Genus	Treatment		SEM	*p*-Value
		CON	MHB		
Bacteroidota	Paraprevotella	0.708	1.333	0.282	0.043
	Alloprevotella	0.028	0.097	0.028	0.029
	Marinilabilia	0.013	0.070	0.026	0.044
Candidatus Saccharibacteria	Saccharibacteria_genera_incertae_sedis	0.110	0.283	0.072	0.030
Bacillota	Blautia	0.150	0.083	0.031	0.048
Verrucomicrobia	Subdivision5_genera_incertae_sedis	0.065	0.113	0.019	0.022
Chloroflexota	Ornatilinea	0.118	0.025	0.021	<0.01

**Table 4 animals-13-03461-t004:** Change characteristics of the main microbial communities in the feces of the MHB diet group.

Phylum	Genus	Treatment		SEM	*p*-Value
		CON	MHB		
Bacteroidota	Prevotella	1.662	3.960	0.998	0.036
Bacillota	Blautia	0.947	0.482	0.213	0.046
	Coprococcus	0.700	0.412	0.118	0.028
	Clostridium XlVb	0.203	0.320	0.050	0.035
	Parasutterella	0.148	0.068	0.024	0.005

**Table 5 animals-13-03461-t005:** Effect of MHB diet on serum antioxidant capacity of fattening sheep.

Item	Treatment		SEM	*p*-Value
	CON	MHB		
GSH-Px	287.419	351.290	22.455	0.012
MDA	2.970	2.094	0.598	0.164
SOD	15.551	20.061	0.639	<0.01
T-AOC	0.288	0.555	0.080	0.004

## Data Availability

The authors confirm that the data supporting the study findings are available in the article and Appendix A.

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
