# Peer review of "Multi-Omics Analysis of the Mechanism of Mentha Haplocalyx Briq on the Growth and Metabolic Regulation of Fattening Sheep"

_animals, 2023, doi:10.3390/ani13223461_

Round 1
Reviewer 1 Report
Comments and Suggestions for Authors
Major issues
-The authors must express clearly and concisely the objectives of the study. As it is now the relevant paragraph, it is difficult for readers to fully understand the aims of the study.
-Please describe in detail the criteria for selecting the animals that were included into this study.
-The description of the techniques in sub-section 2.2. is vague and very limited. Please describe the techniques clearly and fully to allow future readers to be able to replicate the study if needed.
-Sub-section 2.3. All the details of the PCRs please – primers etc. in full detail.
-Sub-section 2.6. Why perform t-test? Were the data normally distributed? How did you prove that? Please explain and please redo the analysis by non-parametric tests if necessary.
Minor issues
-Table 1 can be moved into supplementary material.
-Tables 7, 8 and 9 can be moved into supplementary material.
-The Discussion can be divided into 2 to 4 sub-sections to allow better flow of the reading.
Author Response
Responses to reviewers’ comments
Manuscript ID: animals-2663351
Title: Multi-omics Analysis of the Mechanism of Mentha haplocalyx Briq on the Growth and Metabolic Regulation of Fattening Sheep
Dear Editor and Reviewers:
We are truly grateful to you and the reviewers for the critical comments and thoughtful suggestions on our manuscript. They are really helpful and based on these comments and suggestions, we have revised the manuscript carefully. In the following pages are our point-by-point responses to the reviewers’ comments/suggestions. Please feel free to contact us if there is any question and we are very willing to improve our manuscript until all the reviewers are satisfied.
Best regards,
Hongguo Cao
Address: College of Animal Science and Technology, Anhui Agricultural University, Hefei 230036, P.R. China
E-mail: caohongguo1@ahau.edu.cn
Comment: The authors must express clearly and concisely the objectives of the study. As it is now the relevant paragraph, it is difficult for readers to fully understand the aims of the study.
Answer: Thank you for pointing out the problem. According to your comments and suggestions, we clearly expressed the research objectives in the introduction section of the revised manuscript. (Line72-74)
C: Please describe in detail the criteria for selecting the animals that were included into this study.
A: Thank you for pointing out the problem. According to your comments and suggestions, we explained “criteria for selecting the animals” in detail. (Line84-85)
C: The description of the techniques in sub-section 2.2. is vague and very limited. Please describe the techniques clearly and fully to allow future readers to be able to replicate the study if needed.
A: Thank you for pointing out the problem. According to your comments and suggestions, we further described these technical details in order for readers to better understand the key aspects of the technology. (Line122-150)
C: Sub-section 2.3. All the details of the PCRs please – primers etc. in full detail.
A: Thank you for pointing out the problem. According to your comments and suggestions, we have provided all the information for PCR. (Line160-175)
C: Sub-section 2.6. Why perform t-test? Were the data normally distributed? How did you prove that? Please explain and please redo the analysis by non-parametric tests if necessary.
A: Thank you for pointing out the problem. We are very sorry for our carelessness, we used the SPSSAU data analysis platform to perform a normal distribution test on the data. After confirming that the data conforms to the normal distribution, we conducted an independent sample t-test on the data. (Line208-209)
C: Table 1 can be moved into supplementary material.
A: Thank you for pointing out the problem. According to your suggestion, move Table 1 into the supplementary material.
C: Tables 7, 8 and 9 can be moved into supplementary material.
A: Thank you for pointing out the problem. According to your suggestion, move Table 7,8 and 9 into the supplementary material.
C: The Discussion can be divided into 2 to 4 sub-sections to allow better flow of the reading.
A: Thank you for pointing out the problem. According to your suggestion, the Discussion is divided into four sub-sections.
We tried our best to revise manuscript. These changes will not influence the content and framework of the manuscript. We appreciate for editor’s and reviewers’ critical comments and thoughtful suggestions for our manuscript and hope that the revised manuscript will meet the standard of Animals.
Once again, thank you very much for your comments and suggestions.
Sincerely Yours,
Hongguo Cao

Reviewer 2 Report
Comments and Suggestions for Authors
Expand Conclusion: Take the opportunity to elaborate on the significance of the results and their broader implications. Discuss how the findings contribute to the existing body of knowledge in the field and suggest potential avenues for future research. This study investigates the effects of adding Mentha haplocalyx Briq (MHB) to the diet of meat sheep. The research encompasses various aspects including growth performance, microbiota composition, metabolic profiles, and gene expression. Twelve Hu sheep were studied over a 10-week period, with notable findings.
The positive impacts of MHB supplementation are evident. The average daily weight gain increased significantly by 20.1%, indicating improved growth in the MHB group. Notably, there were alterations in the microbial composition in both the rumen and fecal samples. These changes suggest a potential influence of MHB on the gut microbiome, which is known to impact nutrient utilization and overall health.
Furthermore, alterations in mucosal layer thickness and villi length in different parts of the digestive tract were observed. These changes may indicate improved absorptive capacity, potentially contributing to the enhanced growth performance.
Metabolomic analysis revealed significant variations in metabolite concentrations, particularly in pathways related to protein synthesis and energy metabolism. This suggests that MHB supplementation may influence key metabolic processes crucial for growth.
Transcriptomic analysis highlighted differentially expressed genes associated with immune regulation, energy metabolism, and protein modification. These findings provide insights into the underlying molecular mechanisms influenced by MHB.
However, there are certain areas that warrant consideration. The study lacks a control group receiving a placebo or standard treatment for comparison. Additionally, a larger sample size would enhance the robustness of the findings. Moreover, while the results are promising, further research is needed to validate and expand upon these findings.
In conclusion, this study demonstrates that MHB supplementation positively impacts the growth performance of meat sheep. The comprehensive approach, encompassing microbiota, metabolomics, and transcriptomics, provides a detailed understanding of the mechanisms involved. These findings hold potential implications for optimizing livestock nutrition and performance. Nevertheless, further research is warranted to corroborate these results and explore potential applications in broader agricultural contexts.
Recommendations
- Control Group: Including a control group receiving a placebo or standard treatment would provide a comparative baseline for evaluating the specific effects of MHB. This would strengthen the study's ability to attribute observed changes to MHB supplementation.
- Sample Size: Increasing the sample size would enhance the statistical power and robustness of the findings. A larger sample size would help in generalizing the results to a broader population of meat sheep.
- Duration of the Study: Extending the experimental period could provide additional insights into the long-term effects of MHB supplementation on growth performance, microbiota composition, and metabolic profiles.
- Mechanistic Studies: Conducting further mechanistic studies to elucidate the underlying molecular pathways affected by MHB could provide a more detailed understanding of how it influences growth and metabolism.
- Dietary Analysis: Providing a detailed analysis of the basal diet, including its composition and nutritional content, would offer context for understanding the specific contributions of MHB supplementation.
- Fecal Microbiota Functionality: Assessing the functional capacity of the fecal microbiota through metagenomic or metatranscriptomic analyses could provide insights into the potential metabolic activities influenced by MHB.
- Economic Analysis: Incorporating an economic analysis, such as a cost-benefit assessment, could help evaluate the economic feasibility and practicality of implementing MHB supplementation in livestock production.
- Field Trials: Conducting field trials in commercial farming settings would validate the findings in real-world conditions and provide practical implications for farmers and producers.
- Risk Assessment: Evaluating any potential risks or unintended consequences associated with MHB supplementation, such as possible side effects or interactions with other dietary components.
Author Response
Responses to reviewers’ comments
Manuscript ID: animals-2663351
Title: Multi-omics Analysis of the Mechanism of Mentha haplocalyx Briq on the Growth and Metabolic Regulation of Fattening Sheep
Dear Editor and Reviewers:
We are truly grateful to you and the reviewers for the critical comments and thoughtful suggestions on our manuscript. They are really helpful and based on these comments and suggestions, we have revised the manuscript carefully. In the following pages are our point-by-point responses to the reviewers’ comments/suggestions. Please feel free to contact us if there is any question and we are very willing to improve our manuscript until all the reviewers are satisfied.
Best regards,
Hongguo Cao
Address: College of Animal Science and Technology, Anhui Agricultural University, Hefei 230036, P.R. China
E-mail: caohongguo1@ahau.edu.cn
Comment: Expand Conclusion: Take the opportunity to elaborate on the significance of the results and their broader implications. Discuss how the findings contribute to the existing body of knowledge in the field and suggest potential avenues for future research. This study investigates the effects of adding Mentha haplocalyx Briq (MHB) to the diet of meat sheep. The research encompasses various aspects including growth performance, microbiota composition, metabolic profiles, and gene expression. Twelve Hu sheep were studied over a 10-week period, with notable findings.
The positive impacts of MHB supplementation are evident. The average daily weight gain increased significantly by 20.1%, indicating improved growth in the MHB group. Notably, there were alterations in the microbial composition in both the rumen and fecal samples. These changes suggest a potential influence of MHB on the gut microbiome, which is known to impact nutrient utilization and overall health.
Furthermore, alterations in mucosal layer thickness and villi length in different parts of the digestive tract were observed. These changes may indicate improved absorptive capacity, potentially contributing to the enhanced growth performance.
Metabolomic analysis revealed significant variations in metabolite concentrations, particularly in pathways related to protein synthesis and energy metabolism. This suggests that MHB supplementation may influence key metabolic processes crucial for growth.
Transcriptomic analysis highlighted differentially expressed genes associated with immune regulation, energy metabolism, and protein modification. These findings provide insights into the underlying molecular mechanisms influenced by MHB.
However, there are certain areas that warrant consideration. The study lacks a control group receiving a placebo or standard treatment for comparison. Additionally, a larger sample size would enhance the robustness of the findings. Moreover, while the results are promising, further research is needed to validate and expand upon these findings.
In conclusion, this study demonstrates that MHB supplementation positively impacts the growth performance of meat sheep. The comprehensive approach, encompassing microbiota, metabolomics, and transcriptomics, provides a detailed understanding of the mechanisms involved. These findings hold potential implications for optimizing livestock nutrition and performance. Nevertheless, further research is warranted to corroborate these results and explore potential applications in broader agricultural contexts.
Answer: Thank you for pointing out the problem. According to your comments and suggestions, In the final paragraph of our discussion, we added an explanation of the importance of the research findings and how they can promote healthy breeding. (Line578-585)
C: Control Group: Including a control group receiving a placebo or standard treatment would provide a comparative baseline for evaluating the specific effects of MHB. This would strengthen the study's ability to attribute observed changes to MHB supplementation.
A: Thank you for pointing out the problem. As a forage resource, mint is consumed in small amounts by grazing sheep in actual production. During our experiment, we observed that adding 8% mint to the diet did not have any other effects on fattening sheep. Therefore, whether to add a control group that received a placebo had almost no effect on the experimental results.
C: Sample Size: Increasing the sample size would enhance the statistical power and robustness of the findings. A larger sample size would help in generalizing the results to a broader population of meat sheep.
A: Thank you for pointing out the problem. In the experimental design, it is hoped to determine whether adding 8% mint has beneficial research value and reduce economic losses through a small sample size. In the future, we will conduct in-depth research on the mechanism through a larger sample size and promote the application of the results to healthy breeding.
C: Duration of the Study: Extending the experimental period could provide additional insights into the long-term effects of MHB supplementation on growth performance, microbiota composition, and metabolic profiles.
A: Thank you for pointing out the problem. The general fattening period for sheep is 60 to 70 days. We choose fattening sheep around 4 months old for feeding experiments, and the entire feeding experiment lasts for 10 weeks. After the experiment, fattening sheep aged 6 to 7 months can be put on the market.
C: Mechanistic Studies: Conducting further mechanistic studies to elucidate the underlying molecular pathways affected by MHB could provide a more detailed understanding of how it influences growth and metabolism.
A: Thank you for pointing out the problem. This will be our next research direction. The addition of mint is of great significance for healthy breeding. In the future, we will delve into its potential molecular pathways, explain the specific mechanisms of mint in the process of fattening sheep breeding, and provide insights for its application in animal husbandry.
C: Dietary Analysis: Providing a detailed analysis of the basal diet, including its composition and nutritional content, would offer context for understanding the specific contributions of MHB supplementation.
A: Thank you for pointing out the problem. In Table 1 of the paper, we have listed the feed composition of the CON group and MHB group fattening sheep diets. After you specify the possible shortcomings, I will further revise and improve them carefully. Thank you.
C: Fecal Microbiota Functionality: Assessing the functional capacity of the fecal microbiota through metagenomic or metatranscriptomic analyses could provide insights into the potential metabolic activities influenced by MHB.
A: Thank you for pointing out the problem. We studied the effect of feeding 8% peppermint on the fecal microbiota of fattening sheep, identified significant differences in microorganisms, and discussed them. We apologize for the lack of in-depth analysis. In the future, we will further explore and conduct in-depth research experiments.
C: Economic Analysis: Incorporating an economic analysis, such as a cost-benefit assessment, could help evaluate the economic feasibility and practicality of implementing MHB supplementation in livestock production.
A: Thank you for pointing out the problem. Peppermint is a perennial herbaceous plant widely distributed in the temperate regions of the Northern Hemisphere. It has the characteristics of short growth cycle and high yield. Using 8% peppermint as a substitute for 8% feed can reduce the cost of fattening sheep, generate higher economic benefits, and is highly feasible and easy to promote.
C: Field Trials: Conducting field trials in commercial farming settings would validate the findings in real-world conditions and provide practical implications for farmers and producers.
A: Thank you for pointing out the problem. At the beginning, we hoped to study the impact of mint on fattening sheep through a small sample size, in order to reduce economic costs. After confirming the beneficial effects of feeding mint, we will increase the sample size for further experimental research.
C: Risk Assessment: Evaluating any potential risks or unintended consequences associated with MHB supplementation, such as possible side effects or interactions with other dietary components.
A: Thank you for pointing out the problem. In this experiment, 8% mint was added during the feeding process, and no adverse reactions were observed in the MHB group fattening sheep. In future experiments to explore the mechanism, we will increase the sample size and conduct an in-depth evaluation of whether there are potential issues with feeding 8% peppermint.
We tried our best to revise manuscript. These changes will not influence the content and framework of the manuscript. We appreciate for editor’s and reviewers’ critical comments and thoughtful suggestions for our manuscript and hope that the revised manuscript will meet the standard of Animals.
Once again, thank you very much for your comments and suggestions.
Sincerely Yours,
Hongguo Cao

Round 2
Reviewer 1 Report
Comments and Suggestions for Authors
Before acceptance, the authors must perform a thorough improvement of language throughout the manuscript to correct various linguistic slips.
Comments on the Quality of English LanguageBefore acceptance, the authors must perform a thorough improvement of language throughout the manuscript to correct various linguistic slips.
Reviewer 2 Report
Comments and Suggestions for Authors
no comments